# Effects of Glutamine on Rumen Digestive Enzymes and the Barrier Function of the Ruminal Epithelium in Hu Lambs Fed a High-Concentrate Finishing Diet

**DOI:** 10.3390/ani12233418

**Published:** 2022-12-05

**Authors:** Qiujue Wu, Zhongying Xing, Jiahui Liao, Longlong Zhu, Rongkai Zhang, Saiqiao Wang, Cong Wang, Yan Ma, Yuqin Wang

**Affiliations:** College of Animal Science and Technology, Henan University of Science and Technology, Luoyang 471003, China

**Keywords:** glutamine, high-concentrate diet, lambs, barrier function, ruminal epithelium

## Abstract

**Simple Summary:**

It has been reported that goats fed a high-concentrate finishing diet had increased the short-chain fatty acid production and concentration; a decreased mean ruminal pH, leading to an increase in the epithelium’s permeability; destruction of the tight junctions between cells; and damage to the barrier function. Glutamine, as a conditionally nonessential amino acid, can maintain the acid base balance of body fluids, promote nitrogen balance, and strengthen the mucosa and the integrity of the tight junction proteins. In this study, we aimed to investigate if Gln (Glutamine) would improve the digestive enzymes’ activity, the ruminal epithelial barrier and fermentation, and immune responses by supplying energy to the mononuclear cells, improving the ruminal epithelium’s morphology and integrity, and mediating the mRNA expression of cytokines and tight junction proteins.

**Abstract:**

The present experiment aimed to research the effects of glutamine (Gln) on the digestive and barrier function of the ruminal epithelium in Hu lambs fed a high-concentrate finishing diet containing some soybean meal and cottonseed meal. Thirty healthy 3-month-old male Hu lambs were randomly divided into three treatments. Lambs were fed a high-concentrate diet and supplemented with 0, 0.5, and 1% Gln on diet for 60 days. The experimental results show that the Gln treatment group had lower pepsin and cellulase enzyme activity, propionate acid concentration, and IL-6, TNF-α, claudin-1, and ZO-1 mRNA expression in the ruminal epithelium (*p* < 0.05); as well as increases in lipase enzyme activity, the ratio of propionic acid to acetic acid, the IL-10 content in the plasma, and the mRNA expression of IL-2 and IL-10 in the ruminal epithelium, in contrast to the CON (control group) treatment (*p* < 0.05). Taken together, the findings of this present study support the addition of Gln to improve digestive enzyme activity, the ruminal epithelium’s barrier, and fermentation and immune function by supplying energy to the mononuclear cells, improving the ruminal epithelium’s morphology and integrity, and mediating the mRNA expression of tight junction proteins (TJs) and cytokines.

## 1. Introduction

Rumen function and health are associated with digestive enzyme activity, ruminal pH, ammonia nitrogen and volatile fatty acids (VFA) concentrations, the buffering capacity of the rumen’s content, the composition and proportion of rumen microbes, and the multicellular structure and functional complex of the ruminal epithelium [1,2,3]. Currently, the use of highly concentrated feed to meet the needs of animal growth and production has become mainstream, but this feeding mode is harmful to animals [4]. Quickly fermentable energy sources promote the production rate of acid in the rumen, and lead to a reduction in the ruminal fluid’s pH, an imbalance in the ruminal microflora, impaired epithelial barrier function, digestive dysfunction, hyperosmolarity, and an increase in ruminal toxin concentrations [5,6]; subsequently, toxins, bacteria, and viruses infect the blood from the rumen and ultimately affect the productivity of ruminants [1,7]. It can be avoided under the condition of proper diet management, but it is extremely difficult.

The rumen epithelium of ruminants is an important part of the body to absorb nutrients. It has a wide range of barrier formation characteristics and plays an indispensable role in resisting disease. The results showed that the morphological structure, tight junction (TJ) protein, and inflammatory marker (TNF-α and IFN-γ) mRNA expression of rumen epithelium in high-concentrate fattening sheep changed greatly [4,8], with increased production [9] and concentration [10] of short-chain fatty acids (SCFA), decreased rumen pH [10,11], and an increased concentration of toxins in the rumen fluid [5,12], resulting in the increased permeability of rumen epithelial cells. This destroys the tight junction between cells, weakening the self-healing ability of epithelial tissue and damaging barrier function [13,14]. Therefore, it is crucial to maintain the integrity of the rumen’s epithelial structure, and to realize the potential mechanisms regulating the changes in the ruminal epithelium’s function in animals fed a high-concentrate finishing diet.

Glutamine, as a conditionally nonessential amino acid in the blood and tissues, is extremely important for energy production in cells (enterocytes, renal epithelial cells, and immune cells) and can maintain the acid–base balance of body fluids, promote nitrogen balance, maintain the integrity of the mucosa and tight junction proteins, and promote the proliferation of immune cells. Clinical and animal trials have indicated that the addition of Gln can recover the intestinal immune function and barrier’s function, improve the nutritional metabolism, restore the tight junctions’ integrity, and attenuate enterogenous inflammation [15,16]. Studies in ruminant nutrition have shown that Gln can provide energy for the ruminal epithelium; increase the content of DNA, RNA, and protein in the small intestine mucosa of calves; and increase the growth rate of the small intestine’s mucosal cells [15,17]. Based on the relevant findings regarding Gln in the field of gastrointestinal nutrition in humans, monogastric animals, and ruminants, we can hypothesize that adding Gln into the ruminal diet can reduce the injury of a high-concentrate diet to the rumen epithelial tissue and improve the mechanical mucosal barrier of the damaged rumen’s epithelium to a certain extent. Therefore, the present trial aimed to evaluate the effects of glutamine (Gln) on the activity of digestive enzymes, rumen fermentation, and gene expression related to the regulation of cytokines and tight junction proteins in the rumen epithelium, and to provide insight into the rumen function of Hu lambs fed a high-precision diet.

## 2. Materials and Methods

### 2.1. Glutamine

Pharmaceutical grade Gln (white powder purity > 98.0%) was bought from Henan Honda Biological Medicine Co., Ltd., Luoyang, China, and added to the control diet. The added concentrations in this test were based on the conclusion of Li et al. (2011) [18] and the manufacturer’s proposal.

### 2.2. Lamb, Management, Experimental Diets, and Experimental Design

In total, 303-month-old male Hu sheep with good health and similar weight (26.75 ± 0.65 kg) were stochastically divided into 3 groups with 10sheep in a group, fed as follows: (1) the control diet (CON, a high-concentrate finishing diet containing 40.5% corn, 18% wheat, 5% soybean meal, and 3.45% cottonseed meal); (2) the control diet added with 0.5% Gln (Gln 1); and (3) the control diet added with 1.0% Gln (Gln 2). The experimental period lasted 60 d, including 45 days to collect data and a 15-day adaptation period. Feed offered was adjusted based on body weight during the adaptation period. During the initial 15 d, Hu lambs were adapted to their experimental diets, and the feeding levels were the same as in the experimental period. The control diet was formulated according to the Chinese agricultural industry standard (NY/T 816-2004) ‘Feeding standard for mutton sheep and goats’, and manufactured as complete pellet feed. The feed composition is shown in Table 1. Alfalfa hay and corn straw were crushed into small pieces of 8 mm and mixed into the other feeds for granulation. All the lambs were fed throughout the trial in separate enclosures (4 × 2.0 m^2^) equipped with feed tanks and water during the entire study period. The experimental design and process were authorized by the guidelines of the Institutional Animal Care and Use Committee of Henan University of Science and Technology (approve number: 094-2022).

The lambs were fed twice daily (8:00 and 17:00 h), and the lambs had free access to fresh water during the test. The lambs were fed diets with the same composition and the only difference was the addition of Gln. The Glutamine was sprinkled over the basal diet. The feed provided was calculated based on the last day, and 10% more was taken to reduce the selection of feed ingredients. The amount of feed provided every day and the remaining food at 8:00 were recorded, and the lambs were fed under constant temperature 20 ℃, and experienced day lengths less than 10 h.

### 2.3. Sample Collection

On Day 60, the lambs were fed according to standard feeding procedures and taken to the local abattoir to be slaughtered after a fasting period of 12 h, with 15 lambs (5 per treatment) killed each day. On Day 60, take two blood samples from each lamb; the blood was collected into 2 mL heparinized EP centrifuge tubes through a jugular vein puncture, and plasma was separated. With the addition of anticoagulant blood plasma, after centrifugation (3000× *g* at 4 °C for 15 min), the supernatant was used for detection of the immune index.

According to the ‘Animal Protection Law of the People’s Republic of China’, the lambs were stunned by a rope bolt immediately after blood collection and killed by bloodletting. At least 50 mL of representative rumen fluid samples from the lambs were collected within 30 min, the pH value was measured, and then the samples were filtered with 4 layers of medical Vaseline gauze. Another rumen fluid sample from each lamb was separated into two portions: one for analysis of the lactic acid concentration and enzyme activity, which was stored at −20 °C; the other part was used later for analyses of the short-chain fatty acid (SCFA) concentration, which was stored in 25% (wt/vol) metaphosphoric acid (5 mL rumen fluid:1 mL metaphosphoric acid) at −20 °C.

After the lambs were slaughtered, a section of the rumen wall was collected from the lambs’ abdominal sac within 10 min to bluntly separate muscle layer and serosa layer from rumen epithelium. After separation, it was quickly washed three times in cold PBS and cut into 0.5 × 0.5 cm pieces, and then rapidly frozen in liquid nitrogen. These tissues were used to the extract RNA and protein.

### 2.4. Blood Cytokine Analysis

Commercial ELISA reagents (Blue Gene, Shanghai, China) were used to analyze the cytokines (interleukin-2 (IL-2), interleukin-4 (IL-4), interleukin-6 (IL-6), interleukin-10 (IL-10), interleukin-12 (IL-12), and TNF-α) according to the manufacturer’s instructions. The sensitivity limit of this method was 0.1 g/mL.

### 2.5. Analysis of pH, Digestive Enzyme Activity, and SCFA Concentrations

The collected rumen juice was detected immediately by a pH electrode (model pH B-4; Shanghai Chemical, Shanghai, China) to determine the pH.

Rumen fluid was centrifuged at 4 °C at 2500× *g* for 5 min, then supernatant was crushed by ultrasound for 3 min, and then centrifuged at 13,780× *g* for 5 min. The principle of the enzymatic activity assay was to determine the concentration of the final product after enzymatic reaction under specific substrate conditions. In accordance with manufacturer‘s instructions, we used test reagents (Nanjing Jiancheng Bioengineering Institute, Nanjing, China) to detect the digestive enzymes amylase, pepsin, lipase, and cellulase.

Determination of the volatile fatty acids (VFA) concentration in the ruminal fluid was carried out by gas chromatography (6890 N; Agilent technologies, Avondale, PA, USA) by referring to Wu et al. (2019).

### 2.6. RNA Extraction and qRT-PCR Analysis of Ruminal Wall Samples

A total of 2 μg RNA was extracted from the rumen wall with TRI-zol reagent (Invitrogen Trading (Shanghai) Co., Ltd.), China, and then ground with a homogenizer. Reverse transcription was performed using the manufacturer’s recommended high-volume CDNA reverse transcription kit (Applied Biosystems, Carlsbad, CA, USA). The primer sequences for GAPDH, clodan-1, clodan-4, occludin, ZO-1, IL-2, IL-4, IL-6, IL-10, IL-12, and TNF-α are shown in Table 2.

Real-time quantitative PCR (qRT) was performed with the SYBR^®^Green RT-PCR kit (Merck KGaA, Darmstadt, Germany). The SYBR^®^Green Supermix was used as a qRT-PCR master mixture and each reaction was performed in duplicate. The following PCR amplification parameters were used: 50 °C for 2 min, 95 °C for 20 min, and 95 ℃ for 10 s; the number of cycles was 40. The ratio of absorbance at 260 nm to that at 280 nm was determined. The cycle thresholds for qRT-PCR were recorded and analyzed with 7500 Real-Time PCR software (Applied Biosystems). Relative fold changes at different sites in the spleen were calculated by the 2−ΔΔct method, which was normalized to the mean expression of the above index and gene-specific efficiency [19].

### 2.7. Statistical Analysis

All data obtained from the experiment were expressed as the mean. One-way analysis of variance (ANOVA) was used to evaluate the effect of Gln on the measurement variables. The difference between the processed data was determined by Duncan ‘s multi-interval test (*p* < 0.05). Statistical analysis was performed using SPSS version 21.0 (SPSS Inc., Chicago, IL, USA, 2012).

### 2.8. Ethical Standards

All procedures in this study were approved by the Animal Welfare Committee of Henan University of Science and Technology, P.R of China, and the national guidelines for the care and use of animals (approve number: 094-2022).

## 3. Results

### 3.1. Ruminal Digestive Enzyme Activity

Table 3 demonstrates how dietary Gln supplementation affected the activity of ruminal digestive enzymes in lambs that were fed a high-concentrate finishing diet. The activity of amylase did not differ across the treatments (*p* > 0.05). However, compared to the CON treatment, the Gln treatment notably increased the activity of lipase enzymes and decreased the activity of pepsin and cellulase enzymes (*p* < 0.05), but there was no distinction when these records contrasted with those in the Gln treatment (*p* > 0.05).

### 3.2. Ruminal Fermentation Parameters

Shown in Table 4 are the effects of adding Gln to high concentrate on the rumen fermentation parameters of lambs. No differences were noted in the concentration of lactic acid, acetic acid, butyrate acid, isobutyrate acid, valerate acid, isovalerate acid, and total VFA concentration treatments (*p* > 0.05). However, compared to the CON treatment, the Gln treatment significantly increased the pH and the ratio of acetic acid to propionic acid (*p* < 0.05), but there was no difference compared with the index of the Gln treatment (*p* > 0.05). The propionic acid concentration of the Gln 1 and Gln 2 treatments was lower than that of the CON treatment (*p* < 0.05), and there was no difference in the propionic acid concentration compared to the Gln treatment (*p* > 0.05).

### 3.3. The Concentration of Cytokines in the Plasma

Table 5 shows the effects of dietary-added glutamine on the plasma cytokine concentrations in lambs fattened on a high-concentrate diet. Compared with the CON group, Gln had no effect on the concentrations of IL-2, IL-4, IL-6, and IL-12 in the plasma of lambs fed a high-concentrate diet (*p* > 0.05). The plasma IL-10 concentration in the Gln treatment group was significantly higher than that in the control group; however, there was no significant difference between the Gln treatment groups.

### 3.4. The mRNA Expression of Cytokine Gene in Rumen Epithelium

Table 6 shows the effect of dietary Gln supplementation on the mRNA expression of cytokine genes in the ruminal epithelium of lambs fed a high-concentrate diet. Compared with the CON treatment group, the Gln treatment group did not show a change in the expression of IL-4 and IL-12 mRNA in the ruminal epithelium of the lambs (*p* > 0.05). However, the Gln treatment groups showed significantly increased mRNA expression of IL-2 and IL-10 in the ruminate epithelium of lambs fed a high-concentrate diet (*p* < 0.05). In terms of the mRNA expression of IL-6 and TNF in the ruminal epithelium of lambs fed a high-concentrate diet, the expression levels in the Gln 1 and Gln 2 groups were lower than those in the CON group (*p* < 0.05), and there was no significant difference between the Gln groups (*p* > 0.05).

### 3.5. The mRNA Expression of Tight Junction Protein Genes in the Ruminal Epithelium

Table 7 shows the effects of adding Gln to a high-concentrate finishing diet on the mRNA expression of tight junction protein genes in the ruminal epithelium of lambs. However, compared with the CON treatment, the Gln treatment significantly increased the mRNA expression of claudin-1 in the ruminal epithelium of lambs fed a high-concentrate diet (*p* < 0.05), but there was no significant difference in these indexes between the Gln groups. The Gln 1 and Gln 2 treatments showed a lower mRNA expression of ZO-1 in the ruminal epithelium of lambs fed a high-concentrate diet than the CON treatment (*p* < 0.05), and there was no significant difference in these indexes between the Gln groups (*p* > 0.05).

## 4. Discussion

### 4.1. Effect of Dietary Gln on the Activity of Ruminal Digestive Enzymes of Lambs Fed a High-Concentrate Finishing Diet

In ruminant animals, such as sheep and goats, the rumen’s microorganisms secrete various digestive enzymes (such as proteases, amylases, and pectinases) to degrade and utilize the natural polymers of the feed [7]. The digestive enzymes’ activity is known to affect the activity of ruminal microbes; improve the digestion, absorption, and metabolism of nutrients; and maintain the rumen’s function and health. Studies have confirmed that when feed management is not correct, a high-concentrate fattening diet can reduce the activities of digestive enzymes such as lipase, trypsin, and amylase in ruminants and cause ruminal acidosis [20,21]. In our study, in the CON treatment group, the long-term high-concentration concentrate feeding mode will weaken the activities of pepsin and cellulase, which may be caused by the inactivation of digestive enzymes in Hu sheep caused by high-concentration feed. However, we also confirmed that the activity of rumen protease and cellulase in the Gln treatment was lower, but still better, than that in the CON treatment. These results show that Gln can exert a beneficial effect on the activity of the ruminal digestive enzymes in Hu lambs fed long-term with a highly concentrated diet. In addition, studies have proved that adding glutamine to the diet of non-ruminants can improve the activity of digestive enzymes under both normal and stressed conditions [16,22]. This may be caused by the interaction between Gln and some components of the diet, thus participating in the metabolism and absorption of nutrients. In addition, the increase in digestive enzyme activity may be an explanation for the improvement in the morphology and integrity of rumen epithelium.

### 4.2. Effect of Dietary Gln on the Ruminal Fermentation Parameters of High-Concentrate Finishing Lambs

It has been reported that in the case of poor feeding management, ruminants fed a high-concentrate finishing diet can increase dry matter intake, growth, and milk production in the short term [4], with increased SCFA accumulation [9,10], decreased ruminal pH [10,11], and a higher toxin concentration in the ruminal fluid [5,12], which leads to a risk of subacute ruminal acidosis and dysfunction of the ruminal epithelial barrier. In the current study, the pH value, and the concentrations of lactic acid, acetic acid, butyrate acid, isobutyrate acid, valerate acid, isovalerate acid, and total VFA had no differences, but the pH value was lower. The present study is consistent with that of Penner et al. (2010) [23], which indicated that the rumen can be at risk of subacute ruminal acidosis. However, the addition of Gln in the diet can increase the pH value and the ratio of acetic acid to propionic acid. Our results are almost consistent with those of Penner et al. (2007, 2009) [10,11], which also indicated that Gln can increase starch digestion capacity and feed efficiency. These results may be related to the indirect effect of Gln on the reduction of VFA in the rumen, the metabolism of VFA in the ruminal epithelium, and the balance of the rumen’s microbial flora. In the rumen, the reduction in VFA is mainly realized through the absorption of the ruminal epithelium and the rumen’s flow to the reticulum. The absorption of the ruminal epithelium specifically includes free diffusion and carrier transport. In this study, the decrease in propionic acid content may be related to the accelerated absorption of the ruminal epithelium. Under normal physiological conditions, propionic acid is the primary sugar precursor in ruminants. The body may meet its normal metabolism by accelerating its absorption, resulting in a decrease in the propionic acid concentration in the rumen.

### 4.3. Effects of Dietary Gln on the Concentration and mRNA Expression of Cytokines in Lambs Fed a High-Concentrate Finishing Diet

The cytokines synthesized and secreted by activated immune cells are a class of biologically active molecules. They participate in the activation of inflammation, immune response, tissue repair, and hematopoietic function, and are mediated by mutual regulation and information exchange between immune cells. The effective immune mediators that can promote the proliferation of immune cells are IL-2, IL-4, IL-6, IL-10, and IL-12 [24,25,26]. Parveen and Philip (1997) [27] confirmed that Gln can promote the release of inflammatory cytokines, enhance the proliferation and activation of lymphocytes, and help maintain the immune function of the body. In this experiment, the content of IL-10 in the plasma of lambs fed a high-concentrate diet was significantly increased after Gln was added to the diet. The experimental results are consistent with the conclusions of the studies above [28,29], indicating that the maintenance of immune function may be caused by the enhancement of lymphocyte proliferation and activation by Gln through the promotion of proinflammatory cytokines [30].

In addition, the proliferation of monocytes may be essential to the needs of Gln, and the protection of monocytes is partly due to the dual effects of Gln on the number of cells and cytokines [31]. In addition, the concentrations of IL-2, IL-4, IL-6, and IL-12 in each treatment group were not affected by the addition of Gln to the diet. The difference is that Wells et al. (1999) [32] proved that adding Gln to the diet of mice could promote the production of IL-1b and IL-6 by macrophages in vivo. Caroprese et al. (2012) [29] also found that adding Gln to the diet of dairy cows could improve the production of IL-6 in vivo. The Gln concentration, environmental conditions, and animal species are the possible factors causing these different results. In addition, it is also necessary to consider that there may be congenital differences in the cytokines in the lymphocytes of different animals.

On the other hand, in lambs fed a high-concentrate finishing diet compared with the CON treatment, the Gln treatment significantly increased the mRNA expression of IL-2 and IL-10 in the ruminal epithelium, and decreased the mRNA expression of IL-6 and TNF-α in the ruminal epithelium. These results suggest that Gln may mediate the protection of mucosal surface barrier function in the small intestinal. The beneficial effects of Gln on mucosal immunity might be explained by its antioxidant activity and bacterial translocation. A high-concentrate finishing diet may cause an increase in the ruminal epithelium’s permeability and result in an inflammation response in the ruminal epithelium; however, supplementing the lambs’ diet with Gln reduced the inflammatory reactions or enhanced the immune responses in the ruminal epithelium. These experimental results are consistent with the research of Caroprese et al. (2012) [29]; their experiment showed that Gln can maintain an appropriate level of lymphocyte activation and maintain the normal function of the immune response, so it can be considered a nutrient. In addition, this may be related to the ability of Gln to increase the number of lymphocytes; part of the reason for this is that Gln can reduce the body‘s innate immune system in the inflammatory factor TNF-α, and provide energy for the ruminal epithelium’s lymphocytes and Gln immune-stimulating properties [16,33,34].

### 4.4. Effects of Dietary Gln on the mRNA Expression of Tight Junction Protein Genes in the Ruminal Epithelium of Lambs Fed a High-Concentrate Finishing Diet

TNF- α is an endogenous factor that can change the expression of TJ protein mRNA in the rumen epithelium and plays a significant role in regulating epithelial barrier function. In the present study, we verified that in lambs fed a high-concentrate finishing diet, compared with the CON treatment, Gln treatment significantly increased the mRNA expression of claudin-1 and decreased the mRNA expression of ZO-1 in the ruminal epithelium. These data are basically similar to the research results of Liu et al. (2013) [4], which indicated that the addition of Gln to a high-concentrate finishing diet can provide a protective effect against diet-caused ruminal inflammation; this may be profitable to improve the immune response of Hu lambs and weaken immunosuppression. Some reports indicated that the administration of Gln was found to be beneficial in decreasing intestinal permeability and increasing the mRNA expression of the tight epithelial junction proteins (such as claudin, occluding, and ZO) in humans [35], rats [36], and broiler chickens [16]. These effects may be due to the use of Gln, which is an energy fuel for lymphocytes and epithelial cells and plays an important role in glucose metabolism, while maintaining the acid–base balance and nitrogen balance [15]. At the same time, the rumen’s epithelial barrier is an important component of the ruminant immune system [14]. TJs play an important role in preventing the translocation of toxins and regulating the destruction of the rumen’s epithelial barrier function [4,8]. In our study, the changes in the expression and distribution of TJ proteins and the injury to the epithelial cells of the lambs’ rumen were closely related to the inflammatory response. These results provide a new idea for studying the role of TJ proteins in the immune homeostasis of the ruminal epithelium in ruminants.

## 5. Conclusions

In summary, on the one hand, glutamine can reduce the activity of pepsin and cellulase, the concentration of propionic acid, and the expression of IL-6, TNF-α, CLADIN-1, and ZO-1 in the ruminal epithelium. On the other hand, glutamine can increase lipase activity, the ratio of propionic acid to acetic acid, plasma IL-10 content, and the expression of IL-2 and IL-10. This study shows that dietary Gln may provide energy fuel for mononuclear cells by regulating the expression and secretion of TJs and cytokines, thereby improving the function of the rumen’s epithelial barrier and fermentation, the immune response, and digestive enzyme activity. This study provides new information for the study of the activity of ruminal enzymes, the rumen’s epithelial barrier function, and its relationship with ruminal fermentation, which is helpful to explore the effect of glutamine on ruminal function. Further work is needed to verify the specific mechanisms.

## Figures and Tables

**Table 1 animals-12-03418-t001:** Nutrient level and composition of experimental diet (% DM basis).

Ingredients	Content
Corn straw	25.0
Alfalfa hay	4.2
Corn	40.5
Wheat	18.0
Soybean meal	5.0
Cottonseed meal	3.45
Limestone	1.05
NaCl	0.5
Limestone	0.3
NaHCO_3_	1.0
Premix ^a^ Concentrate/roughage ratio	1.0 7:3
Total	100.0
Nutrient levels	
Net energy (MJ/kg)	12.80
Crude protein	13.25
Ether extract	3.56
Crude fiber	18.50
Acid detergent fiber	13.9
Neutral detergent fiber	25.75
Calcium	0.96
Phosphorus	0.61

Note: ^a^ Premix provided per kg of diet: Cu (from copper sulfate), 7.5 mg; Zinc (bacitracin zinc), 55 mg; Vitamin A (trans-retinoic acid ester), 2500 IU; Vitamin E, 23 IU; Vitamin D (lecithin alcohol), 4000 IU; Fe (from ferrous sulfate), 55 mg; Mn (from manganese sulfate), 25 mg; Co (from cobalt sulfate) 0.3 mg, iodine (from calcium iodate) 0.67 mg; Se (from sodium selenite), 0.3 mg.

**Table 2 animals-12-03418-t002:** Gene-specific sequence primers used in real-time quantitative PCR.

Primer	Sequence (5′→ 3′)	Amplicon Size, bp
Claudin-1	CACCCTTGGCATAGAGTGTA	216
	GACCATAGAAGGAAGCCTGA	
Claudin-4	AAGGGTTACGACCTGCTGTC	238
	GACGTTGTATGCCGTCCGA	
Occludin	GTTCGACCAATGCTCTCTCAG	200
	CAGCTCCCATTAAGGTTCCA	
ZO-1	CGACCGAATCCTCAGGTGAA	163
	AATCACCCACATCGGATTCT	
IL-2	TAGGCCATTACGGCCATGTA	186
	TAGGCCGAGGCGGCCAAAGT	
IL-4	TAGGCCATTACGGCCGGTCA	228
	ACATGGCGGACAATCCATCC	
IL-6	CCAACTTGGGTCTAATACGG	241
	ACCCATCCGTTGTAGGCATG	
IL-10	TTAATGGGTACCTGGGTTGC	239
	CCCTTCTTCGGAGCATATTGA	
IL-12	CGCAGCCTCCTCCTCATA	134
	GCCCTCAGCAGGTTTTGG	
TNF-α	CAGATAAGAAGCCGGTGACC	155
	AGTAGAGCTAAAGCCCTGCA	
GAPDH	GGGTACTCATCTCTACGCCT	180
	GGTCTAAAGTCCCTACCCGA	

**Table 3 animals-12-03418-t003:** Effect of adding Gln to high concentrate on rumen digestive enzyme activity of lambs.

Items	Treatment ^1^			SEM ^2^	*p*-Value ^3^
	CON	Gln1	Gln2		
α-amylase (U/dL)	48.67	50.95	51.68	0.62	0.075
Pepsin (U/mL)	55.62 ^b^	32.34 ^a^	38.19 ^a^	10.64	0.011
Lipase (U/L)	64.29 ^a^	82.67 ^b^	83.49 ^b^	9.85	0.035
Cellulase (U/mL)	450.12 ^b^	310.13 ^a^	376.34 ^a^	60.15	0.027

^1^ CON = control treatment, Gln1 = 0.5% Gln added to the diet on the basis of control treatment; Gln2 = 1.0% Gln added to the diet on the basis of control treatment. ^2^ Standard error of the mean based on pooled estimate of variation. ^3^ There was no significant difference in the value of the common superscript in the same row between a and b when *p* < 0.05; n = 5.

**Table 4 animals-12-03418-t004:** Effects of adding Gln to high concentrate on rumen fermentation parameters of lambs.

Items	Treatment ^1^			SEM ^2^	*p*-Value ^3^
	CON	Gln1	Gln2		
pH	6.13 ^a^	6.38 ^b^	6.39 ^b^	0.07	0.010
Lactic acid (mmol/L)	1.27	1.20	1.22	0.08	0.012
Acetic acid (mmol/L)	55.94	58.67	59.91	1.65	0.034
Propionic acid (mmol/L)	19.62 ^b^	16.54 ^a^	16.30 ^a^	0.57	0.027
Butyrate acid (mmol/L)	15.36	17.06	17.69	0.73	0.058
Isobutyrate acid (mmol/L)	1.96	2.05	2.13	0.12	0.064
Valerate acid (mmol/L)	1.63	1.75	1.81	0.08	0.059
Isovalerate acid (mmol/L)	2.07	2.19	2.31	0.15	0.057
Total VFA (mmol/L)	99.12	102.65	106.04	2.88	0.081
The ratio of acetic of propionate	2.85 ^a^	3.54 ^b^	3.68 ^b^	0.10	0.039

^1^ CON = control treatment, Gln1 = 0.5% Gln added to the diet on the basis of control treatment; Gln2 = 1.0% Gln added to the diet on the basis of control treatment. ^2^ Standard error of the mean based on pooled estimate of variation. ^3^ There was no significant difference in the value of the common superscript in the same row between a and b when *p* < 0.05; n = 5.

**Table 5 animals-12-03418-t005:** Effects of adding Gln to high concentrate on Serum Cytokines and Immunoglobulins of lambs.

Items	Treatment ^1^			SEM ^2^	*p*-Value ^3^
	CON	Gln1	Gln2		
IL-2 (pg/mL)	245.62	327.82	339.51	18.71	0.589
IL-4 (pg/mL)	9.82	10.94	10.88	0.14	0.061
IL-6 (pg/mL)	157.72	138.83	140.18	10.79	0.512
IL-10 (pg/mL)	86.34 ^a^	119.39 ^b^	128.64 ^b^	11.31	0.031
IL-12 (pg/mL)	60.25	69.38	70.32	50.21	0.057

^1^ CON = control treatment, Gln1 = 0.5% Gln added to the diet on the basis of control treatment; Gln2 = 1.0% Gln added to the diet on the basis of control treatment. ^2^ Standard error of the mean based on pooled estimate of variation. ^3^ There was no significant difference in the value of the common superscript in the same row between a and b when *p* < 0.05; n = 5.

**Table 6 animals-12-03418-t006:** Effects of adding Gln to high concentrate on the expression of cytokine gene in rumen epithelium of lambs.

Items	Treatment ^1^			SEM ^2^	*p*-Value ^3^
	CON	Gln1	Gln2		
TNF-α	1.08 ^b^	0.27 ^a^	0.24 ^a^	0.15	0.012
IL-2	1.04 ^a^	1.99 ^b^	2.14 ^b^	0.12	0.021
IL-4	1.02	1.28	1.39	0.16	0.051
IL-6	1.38 ^b^	0.88 ± 0.19 ^a^	0.80 ^a^	0.18	0.013
IL-10	1.10 ^a^	1.68 ± 0.22 ^b^	1.70 ^b^	0.20	0.028
IL-12	1.06	2.68	2.79	0.45	0.052

^1^ CON = control treatment, Gln1 = 0.5% Gln added to the diet on the basis of control treatment; Gln2 = 1.0% Gln added to the diet on the basis of control treatment. ^2^ Standard error of the mean based on pooled estimate of variation. ^3^ There was no significant difference in the value of the common superscript in the same row between a and b when *p* < 0.05; n = 5.

**Table 7 animals-12-03418-t007:** Effects of adding Gln to high concentrate on the expression of tight junction protein gene in rumen epithelium of lambs.

Items	Treatment ^1^			SEM ^2^	*p*-Value ^3^
	CON	Gln1	Gln2		
Claudin-1	1.25 ^a^	2.58 ^b^	2.62 ^b^	0.24	0.026
Claudin-4	1.21	1.35	1.39	0.18	0.058
Occludin	1.17	1.05	1.14	0.21	0.054
ZO-1	1.12 ^b^	0.88 ^a^	0.81 ± 0.04 ^a^	0.03	0.013

^1^ CON = control treatment, Gln1 = 0.5% Gln added to the diet on the basis of control treatment; Gln2 = 1.0% Gln added to the diet on the basis of control treatment. ^2^ Standard error of the mean based on pooled estimate of variation. ^3^ There was no significant difference in the value of the common superscript in the same row between a and b when *p* < 0.05; n = 5.

## Data Availability

The datasets analyzed are not publicly available due to ownership by the funding partners, but are available from the corresponding author on reasonable request.

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
