# Peer review of "Effects of Glutamine on Rumen Digestive Enzymes and the Barrier Function of the Ruminal Epithelium in Hu Lambs Fed a High-Concentrate Finishing Diet"

_animals, 2022, doi:10.3390/ani12233418_

Round 1
Reviewer 1 Report
1. Line 13:e short-chain fatty acid production and concentration,What does e here mean?
2. Line 25: The results showed that the Gln treatment had lower pepsin and cellulase enzyme activity.
3. Line 27 P should be italicized, the same below.
4. Line 70-74 Relevant studies have been carried out on ruminants, so what is the innovation of this research?
5. Line 101 The numbers in Table 1 should be aligned, and it is suggested to mark the ratio of dietary concentrate to forage.
6. Line 106-110 Powdered GIn may be deposited at the bottom of the trough, how to ensure that the lambs get enough Gin?
7. Line 174: -amylase.
8. Line 218: TNF in rumen epithelium of high concentrate lamb (P<0.05)- α.
9. Line 244 Coats→Goats?
10. Line 250-253 This statement contradicts the experimental results, please check it carefully.
11. Line 270-271 I don’t know what it means!
12. Line 304-305 IL-6 has changed, please discuss the experimental results objectively.
13. Line 312 elevated→evaluated?
14. This is just a simple repetition of the results. You should discuss in depth what changes in these indicators might mean? Compared with relevant studies to make your own opinions and interpretations. Line 313-317
15. Line 327: body- α.
16. Line 331-332: Adjustable TNF- α It may change the expression of TJ protein mRNA in rumen epithelium. The expression is not very smooth
17. In Table 7, when the difference between groups is not significant, it is marked, while in other tables, when the difference between groups is not significant, it is not marked, and the format should be unified here.
18. The manuscript should undergo a thorough language check before resubmission.
Author Response
Dear Dear Emilia Yuan, Ph.D. and reviewers: On behalf of my co-authors, we are very grateful to you for giving us an opportunity to revise our manuscript. We appreciate you very much for positive and constructive comments and suggestions on our manuscript (animals-2022909) entitled "Effects of glutamine on rumen digestive enzymes and the barrier function of the ruminal epithelium in Hu lambs fed a high-concentrate finishing diet ".
Those comments are very valuable and helpful for revising our paper and guiding our study. We have studied those comments carefully and have made correction which we hope meet with approval. Revised portion are marked colored in the paper. The following is a point-to-point response to the reviewers' comments and recommendations.
- Line 13:e short-chain fatty acid production and concentration,What does e here mean?
Response: Thank you for your comment. Text errors have been corrected. They are actic acid, acetic acid, butyrate acid, isobutyrate acid, valerate acid, isovalerate acid, and total VFA concentration
- Line 25: The results showed that the Gln treatment had lower pepsin and cellulase enzyme activity.
Response: Thank you for your comment. We have revised in the text as required.
- Line 27 P should be italicized, the same below.
Response: Thank you for your comment. We have revised in the text as required.
- Line 70-74 Relevant studies have been carried out on ruminants, so what is the innovation of this research?
Response: Thank you for your comment. The innovation is to add Gln to animal feed to reduce the damage of long-term intake of high-concentration feed to animal gastrointestinal tract, improve rumen epithelial barrier and fermentation function, and enhance immune response and digestive enzyme activity.
- Line 101 The numbers in Table 1 should be aligned, and it is suggested to mark the ratio of dietary concentrate to forage.
Response: Thank you for your comment. We have revised in the text as required.
- Line 106-110 Powdered GIn may be deposited at the bottom of the trough, how to ensure that the lambs get enough Gin?
Response: Thank you for your comment. Gln is fully mixed with the diet before feeding, stratification is not obvious in a short time
- Line 174: -amylase.
Response: Thank you for your comment. We have revised in the text
- Line 218: TNF in rumen epithelium of high concentrate lamb (P<0.05)- α.
Response: Thank you for your comment. We have revised in the text
- Line 244 Coats→Goats?
Response: Thank you for your comment. We have revised in the text
- Line 250-253 This statement contradicts the experimental results, please check it carefully.
Response: Thank you for your comment. We modified this in the context of the comparison
- Line 270-271 I don’t know what it means!
Response: Thank you for your comment. We have revised in the text
- Line 304-305 IL-6 has changed, please discuss the experimental results objectively.
Response: Thank you for your comment. The changes of IL-6 in different experiments may be related to the Gln concentration, environmental conditions and animal species in other scholars ' laboratories.
- Line 312 elevated→evaluated?
Response: Thank you for your comment. We have revised in the text
- This is just a simple repetition of the results. You should discuss in depth what changes in these indicators might mean? Compared with relevant studies to make your own opinions and interpretations. Line 313-317
Response: Thank you for your comment. We have revised and supplemented some data about these indicators might mean in the text.
- Line 327: body- α.
Response: Thank you for your comment. We have revised in the text
- Line 331-332: Adjustable TNF- α It may change the expression of TJ protein mRNA in rumen epithelium. The expression is not very smooth
Response: Thank you for your comment. We have made changes in the text
- In Table 7, when the difference between groups is not significant, it is marked, while in other tables, when the difference between groups is not significant, it is not marked, and the format should be unified here.
Response: Thank you for your comment. We have revised in the Table 7.
- The manuscript should undergo a thorough language check before resubmission.
Response: Thank you very much. During this revision, the manuscript has been revised via an English speaker by MDPI. If there are still unreasonable places, we will find another professional organization to polish it if necessary.

Reviewer 2 Report
Dear Authors,
I have reviewed your paper entitled ‘‘Effects of glutamine on rumen digestive enzymes and the barrier function of ruminal epithelium in Hu lambs fed high-concentrate finishing diet’’ and have included my comments below.
While the results and premise behind this paper is interesting and worthy of research I feel this paper requires more revisions before being considered for publication. I have detailed line by line my comments below.
A couple of overall comments on the paper. Some more detail is needed overall in the materials and methods section regarding housing and overall management of the lambs as I presently don’t feel like I could repeat the study on the current descriptions. The results section would also benefit from more detailed presentation as I have mentioned in my comments below.
There is a theme throughout the paper discussing ruminal acidosis as if it is an inevitability with concentrate fed lambs which is clearly not the case and needs to be corrected. The term ‘high concentrate’ also needs clarification as this is a subjective term and a term like ‘ad-lib’ would more descriptive if that is correct.
Throughout the paper there are issues with English tenses, wording and incorrectly spelt words. I have highlighted some of these but the paper would benefit from a full check for these issues prior to re-submission. Correcting the language issues throughout may also help with the other issues as it may be a misunderstanding in the way the reader in interpreting it.
Kind Regards,
Detailed comments:
Line 17: Explain abbreviation please. Also remove ‘might’ and re-word sentence to include ‘hypotheses’ or ‘hypothesize’
Line 18: Remove ‘and delete ‘fuel’
Line 29: Explain this abbreviation please
Line 30: Remove ‘and’, replace with ‘,’
Line 31: Remove ‘fuel’
Line 37: Might be a style issue but I strongly feel that the introduction to a scientific paper should not start with ‘’As well known’’
Line 40: Remove ‘etc’
Line 41-42: Sentence needs to be re-worded and reference needed
Line 43: This statement needs to be clarified that tis feeding system can put animals at risk not a certainty of feed management is correct
Line 44-47: Needs to be re worded and sentences shortened. Also while this is technically correct where diets are correctly managed this is not an issue and some clarification on this point needs to be included
Line 51-57: This section needs to be re-worked to make it clearer for the reader.
Line 65: Remove ‘etc’
Line 73: Term ‘high concentration’ needs to be clarified as is subjective
Line 80: A title should not be abbreviated
Line 87: replace ‘and’ with ‘in’. Also a space is needed after ‘3’ and ideally spell out numbers <10 in text
Line 91-92: Details needed on targeted/offered levels of concentrates and also how the adaptation period was handled. A table or some information on the intakes achieved would be very beneficial
Line 106: sentence needs to be re-worded for clarity
Line 107: Very small levels of Gln were offered, giving this should the Gln not have been pre-mixed into the feed to ensure all lambs consumed equal levels? How was equal intake assured?
Line 110: What were the range and means of these values?
Line 114: Randomly taken as in the order the lambs were done? Please clarify.
Line 115: List product used here and what type of heparinized tube
Line 118: Please state what governing body this law is enforced by.
Line 120: Word ‘quickly’ is a subjective term, was this within 30 minutes, 60 minutes? Please clarify.
Line 121: What type of gauze?
Line 137: Tested not used
Line 145: I don’t think this abbreviation has been explained already, please explain if it has not.
Table 3: Are these values SEM’s? P-values should be included on all tables
Line187; Delete ‘the’ between among and all
Line 190-191: Sentence needs to be re-written, hard to understand what the authors are trying to explain here. Also should the P value statement not come after ‘CON treatment’ not before it?
Table 4: Propionic not propionate
Line 199-200: This sentence adds nothing it just restates the title of the paper.
Line 216: All lambs were high concentrate basal diet, so this statement is incorrect as it implies that there is comparison to a low concentrate group
Line 227-228: This sentence adds nothing it just restates the title of the paper.
Line 229: No capitol needed on ‘compared’
Line 233: Should the P value statement not come after ‘CON treatment’ not before it?
Line 244: ‘sheep and coats’ should this be ‘sheep and goats’?
Line 244-245: A reference is required here. Might be a translation/language issue but enzymes is not the correct term here I feel.
Line 247-249: This is when feed management is not correct and this needs to be included in statements such as this
Line 254: ‘Results’ incorrectly spelt
Line 264-268: The authors appear to assume that ruminal acidosis happens with high levels of concentrate feeding regardless of management. This is not true and studies have clearly shown that where managed correctly acidosis is not inevitable and this needs to be clarified and accounted for throughout this paper as currently it reads as every high concentrate animal will eventually get acidosis.
Line 270-271: Sentence is hard to understand and needs to be re-written
Line 282: Third word in sentence is incorrect
Line 290-292: This section would benefit from some referencing
Line 305: Should the word ‘room’ not be ‘group’
Line 322: Syntax error after reference
Line 354: Sentence needs to be reworded.
Author Response
Dear Dear Emilia Yuan, Ph.D. and reviewers: On behalf of my co-authors, we are very grateful to you for giving us an opportunity to revise our manuscript. We appreciate you very much for positive and constructive comments and suggestions on our manuscript (animals-2022909) entitled "Effects of glutamine on rumen digestive enzymes and the barrier function of the ruminal epithelium in Hu lambs fed a high-concentrate finishing diet ".
Those comments are very valuable and helpful for revising our paper and guiding our study. We have studied those comments carefully and have made correction which we hope meet with approval. Revised portion are marked colored in the paper. The following is a point-to-point response to the reviewers' comments and recommendations.
Dear Authors,
I have reviewed your paper entitled ‘‘Effects of glutamine on rumen digestive enzymes and the barrier function of ruminal epithelium in Hu lambs fed high-concentrate finishing diet’’ and have included my comments below.
While the results and premise behind this paper is interesting and worthy of research I feel this paper requires more revisions before being considered for publication. I have detailed line by line my comments below.
A couple of overall comments on the paper. Some more detail is needed overall in the materials and methods section regarding housing and overall management of the lambs as I presently don’t feel like I could repeat the study on the current descriptions. The results section would also benefit from more detailed presentation as I have mentioned in my comments below.
There is a theme throughout the paper discussing ruminal acidosis as if it is an inevitability with concentrate fed lambs which is clearly not the case and needs to be corrected. The term ‘high concentrate’ also needs clarification as this is a subjective term and a term like ‘ad-lib’ would more descriptive if that is correct.
Throughout the paper there are issues with English tenses, wording and incorrectly spelt words. I have highlighted some of these but the paper would benefit from a full check for these issues prior to re-submission. Correcting the language issues throughout may also help with the other issues as it may be a misunderstanding in the way the reader in interpreting it.
Kind Regards,
Detailed comments:
Line 17: Explain abbreviation please. Also remove ‘might’ and re-word sentence to include ‘hypotheses’ or ‘hypothesize’
Response: Thank you for your comment. We have revised in the text as required.
Line 18: Remove ‘and delete ‘fuel’
Response: Thank you for your comment. We have removed.
Line 29: Explain this abbreviation please
Response: Thank you for your comment. We have done it.
Line 30: Remove ‘and’, replace with ‘,’
Response: Thank you for your comment. We have revised in the text as required.
Line 31: Remove ‘fuel’
Response: Thank you for your comment. We have removed.
Line 37: Might be a style issue but I strongly feel that the introduction to a scientific paper should not start with ‘’As well known’’
Response: Thank you for your comment. We have revised in the text as required.
Line 40: Remove ‘etc’
Response: Thank you for your comment. We have done it.
Line 41-42: Sentence needs to be re-worded and reference needed
Response: Thank you for your comment. We have revised in the text as required.
Line 43: This statement needs to be clarified that tis feeding system can put animals at risk not a certainty of feed management is correct
Response: Thank you for your comment. We have revised in the text as required.
Line 44-47: Needs to be re worded and sentences shortened. Also while this is technically correct where diets are correctly managed this is not an issue and some clarification on this point needs to be included
Response: Thank you for your comment. We have revised in the text as required
Line 51-57: This section needs to be re-worked to make it clearer for the reader.
Response: Thank you for your comment. We have done it.
Line 65: Remove ‘etc’
Response: Thank you for your comment. We have done it.
Line 73: Term ‘high concentration’ needs to be clarified as is subjective
Response: Thank you for your comment. We have added it.
Line 80: A title should not be abbreviated
Response: Thank you for your comment. We have revised in the text as required
Line 87: replace ‘and’ with ‘in’. Also a space is needed after ‘3’ and ideally spell out numbers <10 in text
Response: Thank you for your comment. We have revised in the text as required
Line 91-92: Details needed on targeted/offered levels of concentrates and also how the adaptation period was handled. A table or some information on the intakes achieved would be very beneficial
Response: Thank you for your comment. We have added and revised some data in the text as required.
Line 106: sentence needs to be re-worded for clarity
Response: Thank you for your comment. We have revised in the text as required.
Line 107: Very small levels of Gln were offered, giving this should the Gln not have been pre-mixed into the feed to ensure all lambs consumed equal levels? How was equal intake assured?
Response: Thank you for your comment. We have revised in the text as required. The lambs were fed diets with the same composition and the only difference was the addition of Gln. Glutamine is diluted first, then sprayed on the surface of concentrate, and fed after drying
Line 110: What were the range and means of these values?
Response: Thank you for your comment. We have added it.
Line 114: Randomly taken as in the order the lambs were done? Please clarify.
Response: Thank you for your comment. We have revised in the text as required. This word is used incorrectly
Line 115: List product used here and what type of heparinized tube
Response: Thank you for your comment. We have revised and added these data in the text as required.
Line 118: Please state what governing body this law is enforced by.
Response: Thank you for your comment. We have added these data in the text as required.
Line 120: Word ‘quickly’ is a subjective term, was this within 30 minutes, 60 minutes? Please clarify.
Response: Thank you for your comment. We have revised in the text as required.
Line 121: What type of gauze?
Response: Thank you for your comment. We have added the data in the text as required.
Line 137: Tested not used
Response: Thank you for your comment. We have revised in the text as required.
Line 145: I don’t think this abbreviation has been explained already, please explain if it has not.
Response: Thank you for your comment. We have added the data in the text as required.
Table 3: Are these values SEM’s? P-values should be included on all tables
Response: Thank you for your comment. We have added these data in the text as required.
Line187; Delete ‘the’ between among and all
Response: Thank you for your comment. We have revised in the text as required.
Line 190-191: Sentence needs to be re-written, hard to understand what the authors are trying to explain here. Also should the P value statement not come after ‘CON treatment’ not before it?
Response: Thank you for your comment. We have revised in the text as required.
Table 4: Propionic not propionate
Response: Thank you for your comment. We have revised in the text as required.
Line 199-200: This sentence adds nothing it just restates the title of the paper.
Response: Thank you for your comment. We have revised in the text as required.
Line 216: All lambs were high concentrate basal diet, so this statement is incorrect as it implies that there is comparison to a low concentrate group
Response: Thank you for your comment. We have revised in the text as required.
Line 227-228: This sentence adds nothing it just restates the title of the paper.
Response: Thank you for your comment. We have revised in the text as required.
Line 229: No capitol needed on ‘compared’
Response: Thank you for your comment. We have corrected it.
Line 233: Should the P value statement not come after ‘CON treatment’ not before it?
Response: Thank you for your comment. We have corrected it.
Line 244: ‘sheep and coats’ should this be ‘sheep and goats’?
Response: Thank you for your comment. We have corrected it.
Line 244-245: A reference is required here. Might be a translation/language issue but enzymes is not the correct term here I feel.
Response: Thank you for your comment. We have revised and added these data in the text as required.
Line 247-249: This is when feed management is not correct and this needs to be included in statements such as this
Response: Thank you for your comment. We have revised it in the text as required.
Line 254: ‘Results’ incorrectly spelt
Response: Thank you for your comment. We have corrected it.
Line 264-268: The authors appear to assume that ruminal acidosis happens with high levels of concentrate feeding regardless of management. This is not true and studies have clearly shown that where managed correctly acidosis is not inevitable and this needs to be clarified and accounted for throughout this paper as currently it reads as every high concentrate animal will eventually get acidosis.
Response: Thank you for your comment. We have corrected it.
Line 270-271: Sentence is hard to understand and needs to be re-written
Response: Thank you for your comment. We have revised in the text as required.
Line 282: Third word in sentence is incorrect
Response: Thank you for your comment. We have corrected it.
Line 290-292: This section would benefit from some referencing
Response: Thank you for your comment. We have revised in the text as required.
Line 305: Should the word ‘room’ not be ‘group’
Response: Thank you for your comment. We have corrected it.
Line 322: Syntax error after reference
Response: Thank you for your comment. We have revised in the text as required.
Line 354: Sentence needs to be reworded.
Response: Thank you for your comment. We have corrected it.

Round 2
Reviewer 2 Report
I am happy with the corrections and the paper can now proceed in my opinion